# Inhibition of ROS-Scavenging Enzyme System Is a Key Event in Tomato Genetic Resistance against Root-Knot Nematodes

**DOI:** 10.3390/ijms24087324

**Published:** 2023-04-15

**Authors:** Sergio Molinari, Paola Leonetti

**Affiliations:** Bari Unit, Institute for Sustainable Plant Protection IPSP, Department of Biology, Agricultural and Food Sciences, CNR, 70126 Bari, Italy; paola.leonetti@ipsp.cnr.it

**Keywords:** antioxidants, catalase, hydrogen peroxide, pest resistance, RBOHD, ROS, root-knot nematodes, superoxide anions

## Abstract

Genetic resistance in plants against incompatible pests is expressed by the activation of an immune system; however, the molecular mechanisms of pest recognition and expression of immunity, although long the object of investigation, are far from being fully understood. The immune response triggered by the infection of soil-borne parasites, such as root-knot nematodes (RKNs), to incompatible resistant tomato plants was studied and compared to the compatible response that occurred when RKNs attacked susceptible plants. In compatible interactions, the invading nematode juveniles were allowed to fully develop and reproduce, whilst that was impeded in incompatible interactions. In crude root extracts, a first assay of reactive oxygen species (ROS)-scavenging enzymatic activity was carried out at the earliest stages of tomato–RKN incompatible interaction. Membrane-bound and soluble CAT, which is the most active enzyme in hydrogen peroxide (H_2_O_2_) scavenging, was found to be specifically inhibited in roots of inoculated resistant plants until 5 days after inoculation, with respect to uninoculated plants. The expression of genes encoding for antioxidant enzymes, such as CAT and glutathione peroxidase (GPX), was not always inhibited in roots of nematode-infected resistant tomato. Therefore, the biochemical mechanisms of CAT inhibition were further investigated. Two CAT isozymes were characterized by size exclusion HPLC as a tetrameric form with a molecular weight of 220,000 dalton and its subunits (55,000 dalton). Fractions containing such isozymes were tested by their sensitivity to both salicylic acid (SA) and H_2_O_2_. It was evidenced that elevated concentrations of both chemicals led to a partial inactivation of CAT. Elevated concentrations of H_2_O_2_ in incompatible interactions have been suggested to be produced by membrane-bound superoxide anion generating, SOD, and isoperoxidase-enhanced activities. Such partial inactivation of CAT has been depicted as one of the earliest key metabolic events, which is specifically associated with tomato immunity to RKNs. Enhanced ROS production and the inhibition of ROS-scavenging systems have been considered to trigger all the metabolic events leading to cell death and tissue necrosis developed around the head of the invading juveniles by which this special type of plant resistance is exerted.

## 1. Introduction

Plant parasitic nematodes (PPNs) are soil microscopic worms that severely infect almost all crops worldwide causing an annual global crop loss estimated at about USD 80 billion [1]. The most diffuse and damaging PPN families, after entering roots from the soil as mobile juveniles, become sedentary and complete their entire life cycle in the roots (endoparasitic sedentary nematode, ESNs). Co-evolution with ESNs has induced plants to develop immune protein receptors, generated by resistance genes (*R*-genes), which are able to recognize specific nematode effectors. After recognition, an immune response or hypersensitive reaction (HR), often characterized by cell death and tissue necrosis at the site of infection, is triggered against the invading juvenile that is forced to starve or leave the root. Recognition mechanisms and PPN-induced immune responses have recently been extensively reviewed [2,3]. Most of the studies on plant immunity to ESNs have been focused on tomato (*Solanum lycopersicum* L.) carrying the *R-*gene *Mi1.2*, which confers resistance to three species of the ES root-knot nematodes (RKNs) *Meloidogyne* spp. (*M. incognita*, *M. javanica*, *M. arenaria*), as well as to aphids (*Macrosiphum euphorbiae*) and whitefly (*Bemisia tabaci*) [4,5]. Now, it is ascertained that *Mi1.2*-carrying tomato plants respond to RKN infection with a marked and prolonged production of reactive oxygen species (ROS), such as superoxide anion (O_2_^−●^) and hydrogen peroxide (H_2_O_2_). ROS can directly attack the invading juveniles, although they have proven to be signaling molecules for additional immune outputs and responsible of diffuse root cell death, if their cellular level is not controlled [3]. Along with genetic resistance (GR), susceptible plants are able to respond to plant pathogens by an induced resistance, such as systemic acquired resistance (SAR), which shows biochemical mechanisms similar to those observed in GR and is activated by various factors of biotic and abiotic nature, called elicitors, and reflects a certain adaptive potential of the organisms [6].

NADPH oxidases are primarily involved in ROS production as they transfer electrons from cytosolic NADPH to apoplastic oxygen generating O_2_^−●^. By the enzyme superoxide dismutase (SOD), superoxides are rapidly converted into H_2_O_2_, which is as toxic but more stable than O_2_^−●^ and capable of crossing the lipid bilayer of plasma membrane [7]. NADPH oxidases belong to the respiratory burst oxidase homolog (RBOH) family, in which the most highly expressed RBOHD is the component deputed to cell death control, cell-wall-damage-induced lignification, and systemic defense signaling [8]. To avoid uncontrolled peroxidative degeneration, a tightly regulated balance between ROS production and scavenging is functional in cells. Disturbances of this equilibrium lead to oxidative stress to which, normally, plants respond with increased production of anti-oxidant enzymes and non-enzymatic compounds. The generation of a long list of anti-oxidant compounds, such as anthocyanins, flavonols, phenolic acids, ascorbate, and glutathione, as well as the increase in anti-oxidant enzyme activities, such as SOD, ascorbate peroxidase (APX), and catalase (CAT), are favored in response to a ROS augmented cellular level. CAT is a tetrameric heme-containing enzyme found in all aerobic organisms and is very active in converting high concentrations of intracellular H_2_O_2_ in H_2_O and O_2_. The prolonged inhibition of such an activity would lead to an uncontrolled rise in intracellular H_2_O_2_. An event triggered by high H_2_O_2_ in cells is the immune response expressed as HR, cell death and tissue necrosis, which is specifically observed in immunity exerted by *Mi1.2*-carrying tomato plants against RKNs [9].

It has previously been reported that genetic resistance of plants to nematodes is regulated by salicylic acid (SA) using a SA-dependent signaling pathway [10,11,12], as has similarly been found in most incompatible pathogen--plant combinations controlled by resistance genes (*R*-genes) [13]. Salicylic acid (SA) is generally recognized as an inhibitor of both CAT and APX that may facilitate H_2_O_2_ accumulation during the oxidative burst induced by incompatible plant-pathogen/parasite interactions [14]. SA is generated in roots during incompatible tomato–RKN interaction, although it is rapidly transferred to leaves where it markedly induces pathogenesis-related (*PR*) genes as an expression of SAR [11,15,16]. Although the role of SA in triggering hypersensitive cell death seems to be established, its effective inhibition of CAT in immune responses has been questioned [17].

In this paper, we have compared an ROS-scavenging enzyme system and ROS-generating activity in incompatible and compatible RKN–tomato interactions to obtain more insights on the role of ROS in the immunity expressed by *Mi1.2-*carrying tomato plants. Moreover, we have investigated the ability of SA and H_2_O_2_ to inhibit CAT isolated and purified from several cellular fractions of roots of isogenic tomato lines for resistance/susceptibility to RKNs. Immunity expressed as cell death and tissue necrosis has been found to be associated with an activated microsomal ROS production and a contextual early inactivation of CAT.

## 2. Results

### 2.1. Bioassays of Tomato–RKN Interactions

Incompatible tomato–RKN interactions were realized only when *Mi1.2*-carrying resistant tomato cvs were inoculated with an avirulent Meloidogyme incognita population (*MifieldV*). Compatible interactions occurred when *MifieldV* was inoculated on susceptible cvs or a lab-selected virulent population (*SM2V*) was used as inoculum to both resistant and susceptible cvs. In incompatible interactions, J2s were not allowed to develop until reproduction and produced no egg masses; conversely, compatible interactions produced 50–100 egg masses per root system (Figure 1).

### 2.2. ROS-Scavenging Enzyme Assays in Root Crude Extracts Collected from the Earliest Stages of Tomato–RKN Incompatible Interactions

Root tissues were collected from resistant control plants and plants inoculated with *MifieldV* harvested 1 and 2 days after inoculation (dpi). Crude extracts were obtained and used to assay ROS-scavenging enzymes, such as SOD, CAT, and APX (Figure 2). Only CAT was found to be markedly inhibited in roots of plants both 1 and 2 dpi with respect to uninoculated controls. Therefore, it was decided to investigate further the mechanisms of such an inhibition and to test the effect of a successful nematode infection on CAT activity.

### 2.3. CAT Inhibition as a Marker of Tomato Resistance to RKNs

Cytosolic soluble (CFs) and mitochondrial/peroxisomal (MPFs) fractions were isolated from root crude extracts of uninoculated and inoculated plants at 5 dpi. CFs were ultra-filtered as to divide cytosolic proteins from low-molecular-weight compounds, such as soluble phenols (PHE). CAT and PHE were assayed in CFs; membrane-bound CAT and PHE were assayed in MPFs (Table 1). CAT was found to be specifically inhibited only in incompatible tomato–RKN interactions, both in soluble and particulate fractions. On the contrary, PHE markedly increased in particulate fractions but decreased in soluble fractions of roots from resistant inoculated plants, with respect to controls.

In compatible interactions, CAT and PHE did not show significant changes in inoculated plants. Therefore, CAT inhibition and a significant increase in membrane-bound PHE were confirmed to be specific events occurring in tomato roots responding hyper-sensitively to RKNs.

### 2.4. Expression of Antioxidant and Cell Death Promoting Genes Involved in Incompatible and Compatible RKN–Tomato Interactions

Incompatible reactions to avirulent RKN populations in tomato rely on ROS augmented level in cells, probably due to CAT inhibition, as well as on the execution of HR based on the generation of PR proteins [18]; conversely, successful nematode infections have been found to be associated with over-expression of the glutathione peroxidase encoding gene (*GPX*), which is considered a potential detoxifier of H_2_O_2_ [19,20]. Therefore, the number of transcripts of the genes *CAT* and *GPX*, as well as of the hypersensitive cell death inducer *PR4b* gene [21], was recorded by qRT-PCR at 5 dpi in roots of inoculated susceptible and resistant plants and compared with those of uninoculated plants (Figure 3).

Expression of *CAT* and *GPX* is consistently down-regulated in roots of *MifieldV*-inoculated Motelle resistant plants, not markedly affected in VFN8 resistant plants, consistently up-regulated in Moneymaker susceptible plants, with respect to uninoculated plants. The HR-associated *PR4b* gene is highly expressed in incompatible interactions, whilst its expression is inhibited in the compatible interaction. Since high inhibition of *CAT* expression seems not to characterize all incompatible interactions in contrast to the inhibition of the enzyme, further investigation was required to reveal the biochemical mechanisms of such specific inhibition.

### 2.5. Biochemical Mechanisms of CAT Inhibition in Tomato Resistance to RKNs

To mimic nematode infection, plants were provided in pots with SA (20 mg/plant), which is a known inhibitor of CAT. Plants from near-isogenic tomato lines (cvs Motelle/Moneymaker) were harvested 5 days after treatment (dpt). Then, root soluble proteins were recovered and CAT-assayed in untreated and SA-treated plants (Figure 4).

SA treatment of resistant plants induced a significant decrease in CAT in root extracts, similar to that induced by nematode infection. SA treatment to susceptible plants did not apparently change CAT in root extracts.

#### Purification and Characterization of CAT Isozymes from a Tomato Resistant Isoline

To detect if SA directly or indirectly inhibited root CAT in resistant tomato isoline Motelle, a purification of the enzyme was attempted from CF retained protein suspensions. First, CFs were added with ammonium sulphate (AS) to reach 22% (*w*/*v*). Precipitates contained high-molecular-weight proteins; this procedure did not represent a purification step of the enzyme as CAT of the precipitates was similar to that of the starting suspensions. Therefore, 22% AS precipitates were further purified through size exclusion HPLC. Chromatographic eluates were collected after void volumes in 1mL fractions which were analyzed in their absorbance at 280 nm (peak of absorbance of proteins) and 407 nm (peak of absorbance of heme-containing proteins such as catalase); furthermore, total CAT of the fractions between 10th (first after void volume) and 18th (no more CAT detected) was assayed as units mL^−1^ (Figure 5).

The first fraction (10th) of eluate, after void volume, contained the most purified CAT fraction with a specific activity of approx. 30 times higher (about 600 U mg^−1^ prot) than that detected in the loaded sample and a ratio A_407_/A_280_ = 1.3. The molecular weight of this high molecular weight tomato CAT isozyme (hmwtCAT) was calculated as approx. 220,000 dalton, according to a calibration curve designed after loading a calibration kit consisting of β-amylase, alcohol dehydrogenase, bovine serum albumin, ovalbumin, carbonic anhydrase, cytochrome c. HmwtCAT showed a slightly lower elution time than β-amylase (200,000). The 13th fraction contained a CAT isozyme purified 4 times with respect to the loaded sample with a ratio A_407_/A_280_ = 0.8. In this case, the molecular weight was 55,000 dalton (low molecular weight tomato CAT isozyme, lmwtCAT). The presence of this lmwtCAT in the AS precipitate containing high MW proteins can be explained by degradation of the tetramer hmwtCAT, over time and/or during the chromatographic elution, into the subunit lmwtCAT; catalase extracted from chard presented the same tetrameric form [22]. The 22% AS precipitates from CFs were considered to contain mainly the tetramer hmwtCAT.

To recover the subunit lmwtCAT, supernatants from 22% precipitation were added with 90% AS and centrifuged; these second precipitates included low MW proteins, such as lmwtCAT. AS precipitates were obtained from both Motelle and Moneymaker isolines. HmwtCATs (Figure 6A) and lmwtCATs (Figure 6B) were tested in the absence or presence of increasing concentrations of SA (0.1–1.0 mM).

The tetramer hmwtCAT, which should be the main active isozyme under physiological conditions, was the most sensitive to SA in concentrations as high as 0.2 mM, and had a marked inhibitory effect on the enzyme activity. At lower concentrations (0.1 mM), SA acted as an activator or had a negligible inhibitory effect on both the isozymes. It should be noted that standard enzyme assays are performed at a saturating concentration of H_2_O_2_ (20 mM), which is unlikely to be reached even during a hypersensitive reaction to invading pests. Therefore, we arranged enzyme assays to detect the effect of 0.2 mM SA on hmwtCATs, from both resistant and susceptible tomato roots, working at lower H_2_O_2_ concentrations (1–20 mM), which are most probably generated under stress conditions (Figure 7).

It is apparent that, at low H_2_O_2_ concentrations, SA did not produce any inhibiting effect on CAT. Inhibition is maximally exerted when the H_2_O_2_ level drastically increases. Therefore, it can be concluded that the observed specific CAT inhibition in roots of resistant tomato plants in the early stages of nematode infection is probably due to an increase in both H_2_O_2_ and SA.

### 2.6. Generation of H_2_O_2_ in Tomato Resistance to Nematodes

H_2_O_2_ production has its initial step in a superoxide anion (O_2_^−●^) generating membrane-bound NADPH oxidase (NADPH Ox), which was demonstrated for the first time in membrane fraction of fungi-inoculated wounded potato tubers and was reported as RBOHD to have a crucial role in the oxidative burst triggering plant immunity [8,23]. Membrane rich fractions (microsomes) were isolated from root extracts of *MifieldV*-inoculated and uninoculated resistant and susceptible tomato plants. Different enzyme activities were assayed on these fractions, such as the above-mentioned NADPH Ox, SOD, and three isozymes of peroxidase (PEX) using guaiacol (GUA), syringaldazine (SYR) and *p*-phenylenediamine-pyrocathecol (PPD-PC) as substrates (Table 2).

Interestingly, in membrane-rich fraction of roots from inoculated resistant plants, showing an immune response to avirulent RKNs, NAPDH Ox and SOD were found significantly higher than in the corresponding fraction of control roots. The result of such higher enzyme activities together would presumably be a higher rate of H_2_O_2_ production in vivo. Additionally, GUA and SYR isoPEXs were enhanced, probably because of their activity in lignin deposition, although their involvement in NADPH oxidation to produce O_2_^−●^ and H_2_O_2_ cannot be ruled out [24]. Conversely, O_2_^−●^ and H_2_O_2_ generation were apparently restricted in tomato–nematode compatible interaction.

## 3. Discussion

The *Mi1.2*-carrying tomato cultivars used in this study contain intracellular immune receptors (Mi-proteins) that can recognize the presence of effector molecules coming from the three most diffused *Meloidogyne* spp. (*M. incognita*, *M. javanica*, *M. arenaria*) as well as pathogenic insects, such as potato aphids and whiteflies, characterized by the same piercing–sucking feeding habit [5,25]. Against RKNs, immunity is expressed as an HR associated with programmed cell death (PCD) of a localized region around the head of the invading nematode within a few days after infection [9]. Immune response starts while the invading juvenile is tempting to establish a feeding site, after the intercellular migration toward root cortical cells, thus suggesting that cell penetration and effector injection by nematode stylet are required to elicit the response [26]. Probably, the injected nematode effectors promote phosphorylation of some “guard” protein that alters its interaction with the Mi-protein which is activated by the consequent conformational change [27,28]. The activation of R-proteins drives to PCD through the re-programming of gene expression, leading to the synthesis of an array of enzymes, including receptor-like kinases, calcineurin-like phosphoesterases, proteases, one UDP-glucosyl transferase, ATPase, etc. [28,29].

One of the first observable events during the immune response of tomato against RKNs (12 h after infection) is the accumulation of ROS, which is highly sensitive to dipheyliodonium chloride (DPI) inhibition, being DPI a specific inhibitor of O_2_^−●^ generation [30]. O_2_^−●^ generating activity was first found in membrane-rich fractions of potato tubers inoculated with *Phytophtora infestans*. [23]. This activity was characterized as an oxidase probably located on the outer surface of plasma membrane, which transfers electrons from intracellular NADPH to extracellular O_2_ [31]. Such an activity, detected at 5 dpi in this study as NADPH-cytochrome *c* reductase, was found to be enhanced in incompatible tomato–RKN interaction and impaired in infected roots of susceptible plants. The activation of this enzyme depends on protein-kinase activity and external calcium, and more precisely, from Ca^2+^-influx, as proved by a major activation obtained when the Ca^2+^-ionophore A23187 was used [32]. Ca^2+^-dependent regulation of RBOHD during immunity has also been reported [8]. Steady-state Ca^+2^ concentration outside the cells would lower O_2_^−●^ generation to a non-damaging level. Accordingly, the O_2_^−●^ generating activity in microsomes of tomato roots was found to be entirely inhibited by SOD and markedly inhibited by exogenous Ca^2+^ [33]. Chelation of Ca^2+^ in the apoplasm by means of calreticulins (CRTs), which are highly conserved calcium-binding proteins in plants and animals, is one of the strategies used by RKNs to prevent calcium influx into the cells, which may initiate an immune response [2]. Higher O_2_^−●^ generating activity was associated with higher SOD in microsomes of incompatible roots 5 dpi, and this association constitutes a powerful machinery for H_2_O_2_ over-production. H_2_O_2_ accumulation has been found to progressively move from plasma membranes and cell walls to cytoplasm and vacuoles of dying cells of tomato roots undergoing HR after an incompatible attack of *M. incognita* [30]. On the contrary, the enforcement of H_2_O_2_-scavanging systems, such as CAT systemic inducement, which maintains low intracellular H_2_O_2_ levels, is required for successful nematode development [34]. Recently, CAT gene expression was found to be 7 times higher in tomato roots of susceptible RKN-infected plants with respect to uninoculated roots; on the contrary, resistance induced by bio-control agents implied a reduced transcriptional expression of CAT [19]. However, it has been shown herein that *Mi1.2*-carrying tomato cvs can or not react to RKNs by decreasing CAT transcripts; conversely, the inhibition of the enzyme has always been found in roots of challenged resistant tomato plants starting from the earliest stages of nematode invasion. Therefore, we have indirectly proved that CAT should be inhibited in vivo by a rise in both H_2_O_2_ and SA that is specific to incompatible interactions.

CAT involves a two-electron equivalent reduction of H_2_O_2_ to H_2_O by the oxidation of the Ferric basic form of the enzyme (heme Fe^3+^) to Compound I (Fe^5+^); H_2_O_2_ can be oxidized into H_2_O + O_2_ by the reduction of Compound I into the Ferric form (Figure 8A). This catalytic cycle is extraordinarily rapid and makes CAT one of the fastest enzymes known. However, Compound I can be siphoned into the less active ferryl intermediate Compound II by phenols such as SA, which, in turn, can be reconverted into the Ferric form by the same reaction. In the presence of excess H_2_O_2_, Compound II can be oxidized to Compound III, which is inactive and hardly converted back to Compound II (Figure 8B). Therefore, both SA and H_2_O_2_ can inhibit CAT, and H_2_O_2_ may do so irreversibly [35]. Actually, SA has been reported to be involved in *Mi1.2*-mediated resistance of tomato to both RKNs and aphids [10,36].

The question to be confronted is the effective role that SA plays in this type of plant immunity: can SA inhibit catalase in vivo? It has not yet been definitely established whether the ability of SA to inhibit catalase has any real biological significance [37]. It has been reported that in infected tissues, SA levels can approach 0.1 mM [38]. At this concentration, SA has not been found to consistently inhibit tetrameric CAT even at the highest H_2_O_2_ concentration. However, it cannot be ruled out that, locally, SA concentrations can further increase and contribute to the CAT inactivation mediated by high H_2_O_2_ concentrations. Once inactivated, CAT does not recover its previous activity if SA is removed from the reaction, as occurred when SA was discarded from root extracts by ultrafiltration. Actually, CAT in roots of nematode-attacked immune plants seems to be irreversibly inhibited by an excess of O_2_^−●^ and H_2_O_2_. On the other hand, SA is synthesized from trans-cinnamic acid in incompatible plant–pathogen interactions, and this metabolic pathway is stimulated by ROS [12]. Increased ROS level in cells can favor elevated rates of SA generation and, ultimately, lead to inactivation of CAT. The enhanced O_2_^−●^-generating and isoPEX activities, and augmentation of membrane-bound phenols (measured according to SA standards), observed in this study in resistance responses to RKNs, are in agreement with this mechanism of CAT inhibition. Rapid H_2_O_2_ production in excess together with the radical monomers produced by isoPEXs in their peroxidative cycle can explain the necrotic lesions observable in roots at the earliest stages of incompatible plant–RKN interactions [39].

Direct evidence of free SA accumulation in plant immunity is difficult to find because an array of reactions, such as SA glucosylation, methylation and degradation, tend to limit its concentration to a non-toxic level in cells [12,40]. Furthermore, it seems that SA is rapidly transferred from roots to shoots, where it probably acts as an SAR signal in tomato nematode-infected plants [11,16,41]. As shown in this study, phenols such as SA, produced in the immune response of tomato, are transferred from soluble to membrane-rich cell compartments (mitochondria + peroxisomes). Phenol excess would result in an impairment of mitochondrial functions, such as uncoupling and inhibition of mitochondrial electron transport [42], that has since long been confirmed to occur in mitochondria extracted from roots of inoculated resistant tomato plants [43]. The undoubted role of SA in lesion formation and cell death, characteristic of plant immunity to RKNs, may pass through degeneration of mitochondria of cells surrounding the nematode juvenile head.

## 4. Materials and Methods

### 4.1. Procedures to Realize Compatible and Incompatible Tomato–RKN Interactions

Seeds of the tomato susceptible (Moneymaker and Roma VF cvs) and resistant (Motelle and VNF8 cvs) to root-knot nematodes (RKNs) were germinated in sterilized quartz sand at 23–25 °C in a glasshouse. Seedlings were transplanted to 110 cm^3^ clay pots filled with 150 g of sterilized river sand and randomly located in temperature-controlled benches (soil temperature 25 ± 2 °C). Seedlings were let to grow, with a regular regime of 12 h light/day and watered with Hoagland’s solution (5 mM KNO_3_, 5 mM Ca(NO_3_)_2_, 1 mM KH_2_PO_4_, 1 mL L^-1^ of a micronutrient stock solution (2.86 g boric acid, 1.81 g MnCl_2_*●* 4H_2_O, 0.22 g ZnSO_4_*●*7H_2_O, CuSO_4_*●*5H_2_O, 0.025 g NaMoO_4_ per liter), and traces of FeCl_3_, pH 6.5 with KOH), for the time needed to reach the 4–6 compound leaf stage and a fresh weight of 4–5 g. In some experiments, plants were soil-drenched with 20 mg/plant K-salicylate (K-SA, pH 6.0), to test the effect of SA on the enzyme catalase extracted from roots.

Two populations of *Meloidogyne incognita* (Kofoid et White) Chitwood 1949 were used: one avirulent population collected in a field and reared on susceptible tomato in a glasshouse (*MifieldV*), and one virulent population artificially selected for resistance breaking by mass inoculation on resistant cvs in a glasshouse (*SM2V*). Incompatible interactions were realized by inoculations of tomato cvs carrying the resistance gene *Mi-1.2* with *MifieldV*; conversely, compatible interactions were obtained by inoculations of tomato cvs carrying the resistance gene *Mi-1.2* with *SM2V* or by inoculations of susceptible cvs with *MifieldV*.

### 4.2. Nematode Inoculations and Infection Level Determination

Invasive second-stage juveniles (J2s) of *MifieldV* and *SM2V* were obtained by incubation of egg masses in tap water at 25 °C; both susceptible and resistant plants were inoculated with 250 J2s/plant by pouring 1–2 mL of J2 stirring suspensions into 2 holes made in the sand at the base of plants. Plants were harvested 40 dpi. Then, roots were cut from shoots and washed free of soil debris. Infection was measured as the number of egg masses per root system (EMs). EMs were red-colored by immersion of root samples in 0.1 g L^−1^ Eosin Yellow, which were stored in a refrigerator for at least 1 h. Root samples were scored for red-colored EMs under a stereoscope (6× magnification).

### 4.3. Extraction of Cellular Fractions from Root Samples

Root samples from uninoculated and inoculated plants were collected at 1, 2, and 5 dpi. Cellular fractions were obtained as follows: roots were separated from shoots and kept in an ice bath. Fresh or stored roots were immersed in liquid nitrogen and ground in porcelain mortars kept in ice baths. Ground powders were stored at −80 °C or immediately suspended in a grinding buffer (1:5, *w*:*v*) of 0.1 M potassium phosphate buffer (pH 6.0), added with 4% polyvinylpyrrolidone and the protease inhibitor phenyl-methane-sulfonyl fluoride (PMSF, 1 mM).

Suspensions were then ground using a Polytron^®^ PT–10–35 (Kinematica GmbH, Malters, Switzerland), and filtered through four layers of gauze. These first crude extracts from samples collected at 1 and 2 dpi were filtered through 0.45 µm nitrocellulose filters applied to 10-mL syringes and used to assay the activities of the ROS-scavenging enzymes SOD, CAT, and APX. Crude extracts from samples collected at 5 dpi were processed further and subjected to a centrifugation for 15 min at 12,000× *g*; pellets were used as Mitochondrial/Peroxisomal Fractions (MPFs) for the determination of phenols and catalase activity. Operations on supernatants followed two different procedures. One part of the supernatants was spun at 100,000 *g* for 90 min in a Bechman ultracentrifuge to obtain microsomal fractions (MicroFs) to be used for the determination of several enzyme activities. Another part of the supernatants was filtered through 0.45 µm nitrocellulose filters applied to 10 mL syringes; filtrates were ultra-filtered at 4 °C through 50 mL Vivaspin micro-concentrators (10,000 molecular weight cut off, Sartorius Stedim, Biotech GmbH, Berlin, Germany). Unfiltered fractions (retentates), containing most of the soluble proteins of the extracts, were used to detect protein content and CAT. Ultra-filtrates were collected and used for phenols detection.

### 4.4. Purification of CAT from Soluble Fractions of Root Samples

The purification of CAT from root soluble fractions of uninoculated susceptible and resistant tomato plants was carried out to study the sensitivity of the enzyme to K-SA and H_2_O_2_. After grinding roots in 0.1 M K-Pi (pH 6.0) added with 10% glycerol and 10 mM di-thiothreitol (1:5, *w*:*v*), homogenates were centrifuged at 9000× *g* for 15 min. Supernatants were added with 22% ammonium sulphate and stirred overnight in a refrigerator (4 °C) to obtain the precipitation of high-molecular weight proteins; suspensions were centrifuged at 14,000× *g* for 25 min. Pellets were dissolved and stored at 4 °C to be used for CAT detection (hmwtCAT) and further CAT purification. Supernatants were brought to 90% ammonium sulphate to precipitate the fraction containing low-molecular-weight proteins. Pellets were obtained by centrifugation at 9000× *g* for 15 min and suspended in the extraction buffer modified with 25% glycerol and 1 mM di-thiothreitol. Suspensions were mixed with 0.8 volumes of an ethanol/chloroform (3:1) organic mixture and 1 mM PMSF, and kept in an ice bath. Water phases were collected and centrifuged at 47,000× *g* for 25 min. Supernatants were used to determine CAT (lmwtCAT). Therefore, fractions of both high- and low-molecular-weight proteins were used to determine the sensitivity of isoCATs to increasing concentrations (0.1–1 mM) of K-SA solutions. Conversely, precipitates containing high molecular weight proteins were used to assay hmwtCAT in presence of increasing H_2_O_2_ concentrations.

To determine the exact molecular weight of hmwtCAT, protein fraction precipitated at 22% ammonium sulphate from root extracts of Motelle tomato cv was subjected to a chromatographic purification step. Aliquots of protein suspensions were filtered through 0.45 µm Whatman filters and analyzed by size exclusion HPLC through a 300 × 7.8 mm BIOSEP SEC-S3000 column (phenomenex^®^). The column was equilibrated with 50 mM phosphate buffer, pH 7.6, plus 0.15 M NaCl at a flow rate of 0.5 mL min^−1^. Proteins (approx. 0.6 mg) were injected in each run. A detector was set at 280 and 407 nm by a dual-channel signal mode and 1 mL fractions were collected 15 min after the injection. High CAT was detected starting from the 10th fraction; no CAT was detected after the 17th fraction.

### 4.5. Proteins, Phenols, and Enzyme Spectrophotometric Assays

Protein and phenol contents were determined by the enhanced alkaline copper protein assay using Folin phenol reagent as colorant [44]; solutions of bovine serum albumin (10–100 µg) or K-salicylate (5–20 µg) were used as calibration curves for protein and phenol content calculation, respectively. These types of assays were carried out by a Beckman DU 70 UV/visible spectrophotometer.

The initial rate of disappearance of hydrogen peroxide was determined to calculate CAT of the different root protein fractions [45]. Reaction mixtures (0.5 mL final volume) consisted of 20 mM H_2_O_2_ and 10–50 µL root extracts in 0.1 M sodium phosphate, pH 7.0; H_2_O_2_ oxidation was monitored as decrease in the absorbance at 240 nm, and one unit of enzyme represented the oxidation of 1 µmole H_2_O_2_ per min (ε = 0.038 mM^−1^ cm^−1^). SOD activity in crude extracts and microsome fractions was measured as the percentage of inhibition of cytochrome *c* (20 µM) reduction by the xanthine-xanthine oxidase system. The xanthine-xanthine oxidase system produces superoxide anions (O_2_^−●^), which cause cyt*c* reduction and increase in the ratio A_550_/A_540_; SOD inhibits the reduction by subtracting O_2_^−●^ (forming H_2_O_2_) to the reaction. Reaction mixtures (1 mL) consisted of 50–100 µL of crude extract or 10–20 µL microsome suspension, xanthine (1 mM), cytochrome *c*, and 0.5 mM EDTA in 0.1 M Na-K-phosphate buffer (pH 7.8). Xanthine oxidase (20 mU) addition started the reaction and cytochrome *c* reduction was monitored at 550/540 nm; one unit of enzyme produced 50% inhibition in cyt*c* reduction with respect to the standard reaction [46]. APX was determined as the rate of disappearance of ascorbate in presence of hydrogen peroxide [47]. The reaction mixture (0.5 ml final volume) contained 0.1 M TES, pH 7.0, 0.1 mM EDTA, 1 mM ascorbate, 0.1 mM H_2_O_2_, 10–20 µL crude extracts. Ascorbate oxidation was monitored by decrease in absorbance at 298 nm in a double-beam spectrophotometer (PerkinElmer 557); one Unit of enzyme expressed the oxidation of 1 µmole ascorbate per min (ε = 0.8 mM^−1^ cm^−1^).

MicroFs were isolated and used for determination of NADPH-cytochrome *c* reductase activity (O_2_^−●^ generating activity), SOD, and guaiacol, syrigaldazine, *p*-phenylenediamine-pyrocatechol isoperoxidases. NADPH-cytochrome *c* reductase activity was assayed by monitoring at room temperature the increase in absorbance at 550 nm, with the reference wavelength set at 540 nm, due to the reduction in externally added oxidized cytochrome *c* (20 µM, horse heart, Sigma type III) by NADPH (10 µM). Reaction mixtures (1 mL final volume) consisted of 0.1 M potassium phosphate buffer, pH 7.8, approx. 10–20 µL microsome suspension and 20 mM NaN_3_; NaN_3_ was used to inhibit cytochrome *c* oxidase. The reduction in cytochrome *c* in the absence of appropriate amounts of such an inhibitor was non-linear and slower. Each measurement of the enzyme activity, expressed as nmoles cyt. *c* reduced min^−1^ mg^−1^ protein, with a ε_550/540_= 19 mM^−1^ cm^−1^. SOD (from bovine erythrocytes, Sigma Co, Milan, Italy.) was added at the concentration of 15 µg ml^−1^ to evaluate the SOD-sensitive, O_2_^−●^ generating activity of the samples. SOD completely inhibited cyt*c* reduction, thus attesting the specific O_2_^−●^ generating activity of NAPDH oxidation in tomato microsomes.

Guaiacol, syringaldazine, and PPD-PC oxidations were monitored at the absorbance of 470, 530, and 557 nm, respectively. Reaction volumes were 1 mL of assay mixtures containing:-An amount of 0.05 M phosphate buffer (pH 6.0), 5 mM guaiacol or 50 µM syringaldazine, 2 mM H_2_O_2_, and 10–20 µL microsome suspension.-An amount of 0.1 M Tris-HC1 buffer (pH 7.6), 0.35 mM PPD/4.5 mM PC, 2 mM H_2_O_2_, and 10–20 µL microsome suspension.-One unit of isoperoxidase activity indicated difference in absorbance in Arbitrary Units (∆AUx) min^−1^ mg^−1^ prot.

All assays of enzyme activity were carried out by a double-beam spectrophotometer (Perkin Elmer 557).

### 4.6. RNA Extraction, cDNA Synthesis, and Quantitative Real-Time Polymerase Chain Reaction

Roots from susceptible and resistant plants inoculated with *MifieldV* were collected at 5 dpi. Samples of roots were weighed and ground to a fine powder in liquid nitrogen. Total RNA extractions were carried out using aliquots of powdered samples (100 mg) by RNA-easy Plant Mini Kits (Qiagen, Germany), according to the instructions specified by the manufacturer. Electrophoresis runs on 1.0% agarose gels were used to test RNAs quality; RNAs were quantified by means of a Nano-drop spectrophotometer. Total RNAs (1 μg samples) were converted into cDNAs by QuantiTect Reverse Transcripton Kit (Qiagen, Germany) with random hexamers, according to the manufacturer’s instructions. PCR mixtures (20 μL final volume) contained RNAse free water, 0.2 μM each of forward and reverse primers, 1.5 μL cDNA template and 10 μL SYBR^®^ Select Master Mix (Applied Biosystems, Italy). PCR cycles consisted of an initial denaturation step at 95 °C (10 min); 40 cycles at 95 °C (30 s), at 58 °C (30 s), at 72 °C (30 s), with a final extension step at 60 °C (1 min). qRT-PCRs were performed in triplicate using an Applied Biosystems^®^ StepOne™ instrument. The following tomato genes were tested: glutathione peroxidase (XM_004244468.3, *GPX*), catalase (NM_001247257.2, *CAT2*), and pathogenesis-related gene 4b (NM_001247154.1, *PR-4b*). For each oligonucleotide set, a no-template water control was used. Actin-7 (NM_001308447.1, *ACT-7*) was used as the reference gene for quantification, as it was experimented to be the most suitable one for the experimental conditions used in this work. Primers were:F: GTTTGCTTGCACACGGTTTA/R: CGTCGTTGGTGGATACCTCT for *GPX*,F:TGCTCCAAAGTGTGCTCATC/R:TTGCATCCTCCTCTGAAACC for *CAT 2*F: TGACCAACACAGGAACAGGA/R: GCCCAATCCATTAGTGTCCA for *PR-4b*F: CAGCAGATGTGGATCTCAAA/R: CTGTGGACAATGGAAGGAC for *ACT-7*

The threshold cycle numbers (Ct) for each transcript quantification were examined and the relative fold changes in gene expression between inoculated and uninoculated roots were calculated by the 2^−∆∆CT^ method [48].

### 4.7. Experimental Design and Statistical Analysis

Cell fractions and root extracts from non inoculated and nematode inoculated plants were obtained from 3 different biological assays. All the corresponding values of enzyme activity and protein and phenol contents are shown as means (*n* = 9) ± standard deviation (SD), considering that each protein sample were tested thrice at the spectrophotometer. Means of values from non inoculated plants were used as controls and differentiated from those from inoculated plants by means of a *t-*test (Microsoft Excel). Asterisks indicate significant difference (* *p* < 0.05). Enzyme purification from roots of healthy resistant and susceptible plants was carried out from one large batch of plants at the same growth stage as plants that were nematode inoculated. *CAT* of purified fractions are means (*n* = 3) ± standard deviation obtained by 3 replicates of runs at spectrophotometer.

As concerns qRT-PCR, RNA was extracted from plants coming from 2 independent bioassays; roots from 2 plants of the same bunch constituted one sample; qRT-PCR runs were performed in triplicate for samples coming from each independent bioassay. qRT-PCR data are expressed as means (*n* = 6) ± standard deviations of 2^−∆∆Ct^ each group from inoculated plants, considering as 1 the values of each group from not inoculated plants, taken as controls; significant difference with respect to controls was determined by the non-parametric Kolmogorov–Smirnov test * *p* < 0.05; ** *p* < 0.01).

## 5. Conclusions

In this study, evidence has been brought indicating that one of the earliest key events in tomato immunity to RKNs is the partial inactivation of CAT by an excess level of its own substrate (H_2_O_2_). A specific accelerated synthesis of O_2_^−●^ causes, after SOD dismutation, an elevated level of H_2_O_2_ that ultimately stimulates a faster synthesis of SA. On the other hand, ROS production can commonly be observed, associated with every type of plant abiotic and biotic stress, wounding included. However, this unspecific ROS production belongs to the so-called phase I of the oxidative burst, which is described as relatively short-lived and occurring after the addition of either compatible or incompatible pathovars of *Pseudomonas syringae* to plant suspension cells [49]. Phase II, depicted as the earliest detectable reaction of plant cells to incompatible pathogens, is a relatively long-lived and more active response after pathogen recognition. Accordingly, these two phases of ROS production have already been described for compatible and incompatible reactions of tomato to RKNs [30]. The prolonged phase II of ROS production, which may be genetically determined after pathogen/pest recognition, seems to be responsible for CAT inhibition. On the contrary, chemical or mechanical injury of tomato did not result in CAT inhibition [50]. These events trigger an HR that ultimately leads to cell death and tissue necrosis, observable around the head of the nematode juveniles, probably because of the collapse of vacuole membranes and liberation of lytic enzymes and/or mitochondrial degeneration. Consequently, juveniles starve to death or are forced to move out of the roots. Cell degeneration would diffuse signal molecules to root distal cells and then to shoot and leaves; the methylated volatile form of SA, methyl-salicylate, is likely to transport such a signal. Interestingly, HR may not be required for *Mi1.2*-mediated resistance [29]. In an alternative circumstance, immunity may be expressed as augment of phenylpropanoid pathway rate and higher production of anthelmintic compounds, such as coumarins, stilbene and isoflavonoid phytoalexins, as well as cell defensive molecules, such as chalcone and lignin [7,51]. The depletion of NADPH because of the augmented rate of NAPDH Ox would stimulate the breakdown of carbohydrates through the pentose phosphate pathway, which is important for the biosynthesis of the aromatic rings system that, via shikimic acid pathway, leads to chorismate, L-arogenate and phenyl-alanine, from which the phenylpropanoid pathway starts [52].

## Figures and Tables

**Figure 1 ijms-24-07324-f001:**
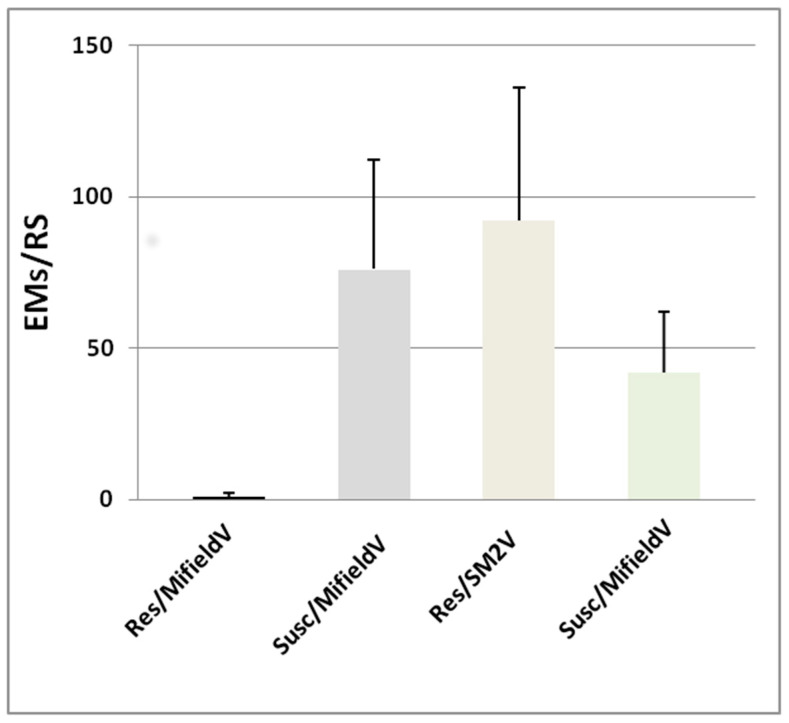
Numbers of egg masses per root system (EMs/RS) in tomato resistant (Res) and susceptible (Susc) plants inoculated with an avirulent (*MifieldV*) and a lab-selected virulent (*SM2V*) *M. incognita* population.

**Figure 2 ijms-24-07324-f002:**
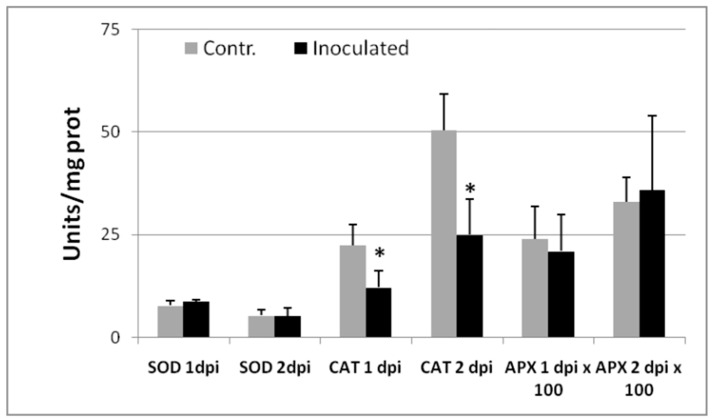
SOD, CAT, and APX activities in root crude extracts from control and resistant plants inoculated with *MifieldV*. Crude extracts were collected from plants harvested 1 and 2 days after inoculation (dpi). Means of enzyme activities are expressed as Units × mg^−1^ prot ± standard deviation (SD) and separated by a paired *t*-test (* *p* < 0.01).

**Figure 3 ijms-24-07324-f003:**
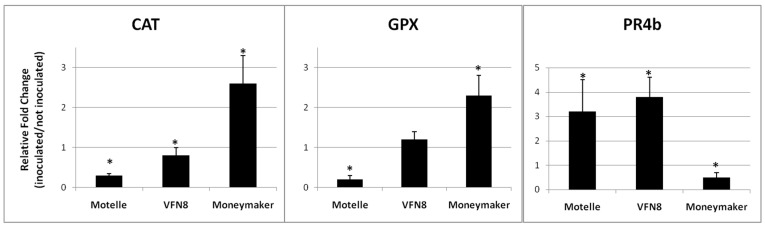
Expression of *CAT*, *GPX*, and *PR4b* genes in roots of tomato plants from the resistant cvsMotelle and VFN8 and the susceptible cv Moneymaker detected by quantitative real-time reverse-transcription polymerase chain reaction (qRT-PCR). Relative fold changes (the value 1 indicates no change) of nematode inoculated with respect to uninoculated plants were measured at 5 dpi. Data are the mean fold changes (*n*= 6) ± SD in gene transcript levels. An asterisk indicates that the mean fold change is significantly different from 1 as determined by the non-parametric Kolmogorov–Smirnov test (*p* < 0.05).

**Figure 4 ijms-24-07324-f004:**
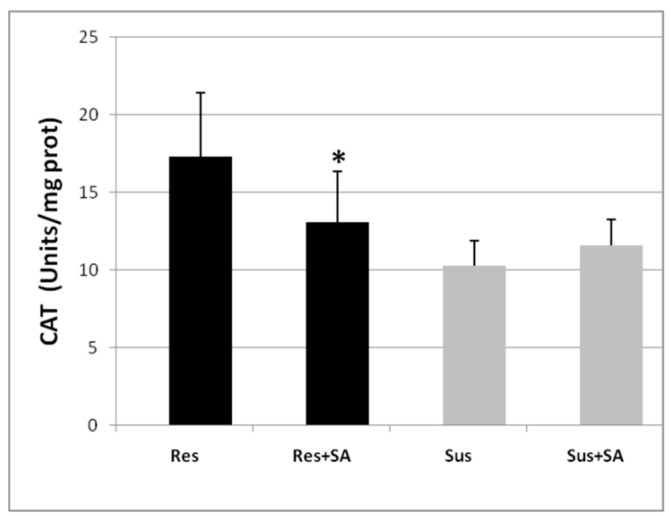
CAT in root extracts of tomato plants from near-isogenic lines resistant (Res, cv. Motelle) and susceptible (Sus, cv. Moneymaker) untreated or treated with SA (20 mg/plant); data were taken at 5 dpt. Values are expressed as means (*n* = 9) ± SD. An asterisk indicates significant difference with respect to controls (Res, Sus), according to a paired *t*-test (*p* < 0.05).

**Figure 5 ijms-24-07324-f005:**
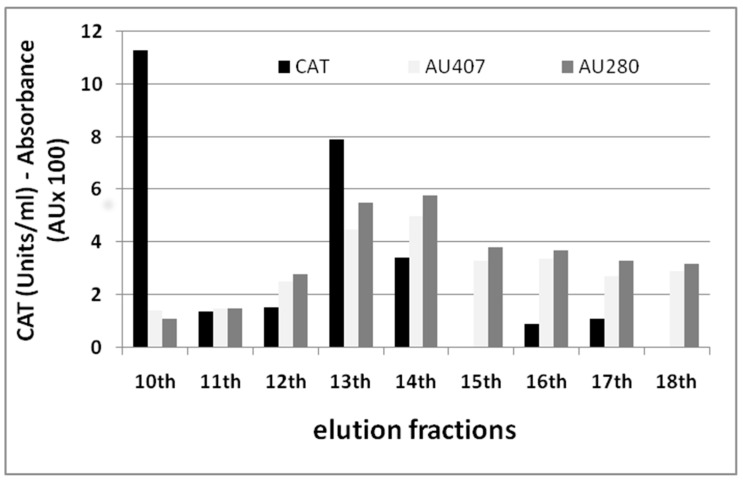
CAT purification from 22% ammonium sulphate precipitates of root cytosolic fractions isolated from resistant tomato cv. Motelle. Precipitates were loaded on a size exclusion column inserted in an automated HPLC system. After void volumes (up to 9th fraction), 1 mL fractions of eluate were collected until they showed detectable CAT, from 10th to 18th fraction. Absorbance of such fractions was measured at 280 and 407 nm, and expressed as arbitrary units (AU) × 100. CAT was expressed as units mL^−1^.

**Figure 6 ijms-24-07324-f006:**
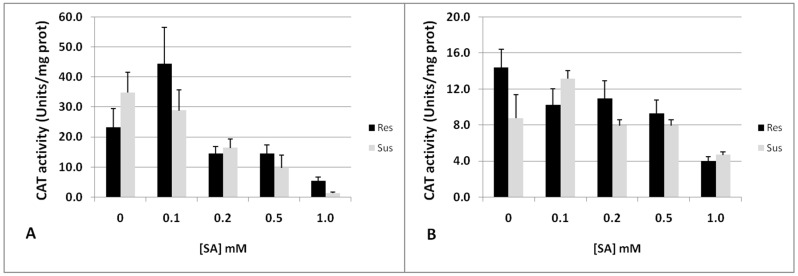
High- (**A**) and low- (**B**) molecular-weight isoCATs (hmwtCAT and lmwtCAT, respectively) activity of AS-precipitates from root soluble fractions of tomato plants resistant (Motelle, Res) and susceptible (Moneymaker, Sus) to RKNs. CAT was assayed in absence and presence of increasing concentration of K-salicylate (0.1–1.0 mM, pH 6.0). Values are expressed as means (*n* = 3) ± SD.

**Figure 7 ijms-24-07324-f007:**
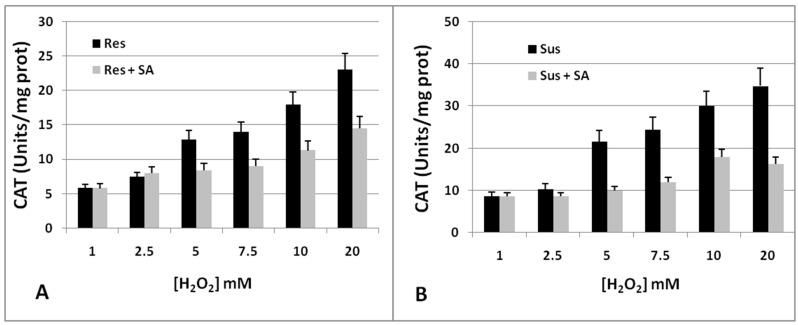
HmwtCATs measured as units mg^−1^ prot from resistant (Motelle, Res, (**A**)) and susceptible (Moneymaker, Sus, (**B**)) tomato roots at increasing H_2_O_2_ concentrations in presence of 0.2 mM K-salicylate (SA). Values are expressed as means (*n* = 3) ± SD.

**Figure 8 ijms-24-07324-f008:**
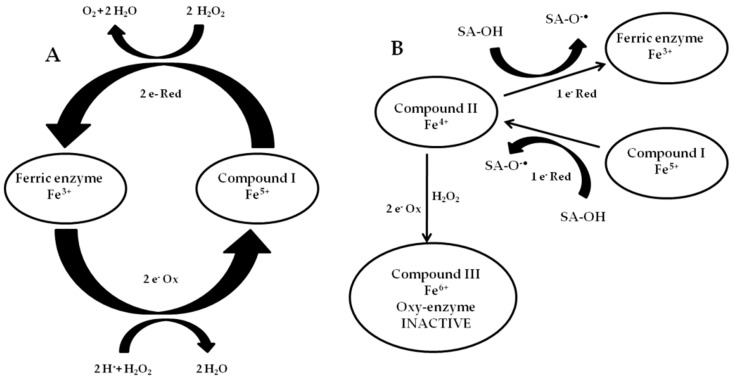
Catalytic cycles of CAT. (**A**) Two-electron equivalent transfer from Ferric enzyme (heme iron positive charges in ovals) to Compound I and vice versa; (**B**) One-electron reduction from Compound I to Ferric enzyme through Compound II; Compound II can be oxidized by H_2_O_2_ into the inactive oxy-enzyme Compound III.

**Table 1 ijms-24-07324-t001:** Catalase activity (CAT, units/mg prot) and phenol content (PHE, µg/g rfw) in root cytosolic fractions (CFs) and mitochondrial/peroxisomal fractions (MPFs). Resistant (cvsMotelle and VFN8) and susceptible (cvs Moneymaker and Roma VF) tomato plants were inoculated with a *M. incognita* avirulent population (*MifieldV*) or left not inoculated as a control (cntr). Motelle cv. was inoculated also with a lab-selected *M. incognita* virulent population (*SM2V*) to have another compatible interaction. Data from incompatible and compatible RKN–tomato interactions were determined at 5 dpi. Values are expressed as means (*n* = 9) ± SD. Asterisks indicate significant difference with respect to controls according to a paired *t*-test (*p* < 0.05).

	Incompatible Interactions				Compatible Interactions					
	cv. Motelle		cv. VFN8		cv. Moneymaker		cv. Roma VF		cv. VFN8	
	cntr	+ *MifieldV*	cntr	+ *MifieldV*	cntr	+ *MifieldV*	cntr	+ *MifieldV*	cntr	+ *SM2V*
CFs										
CAT	27.4 ± 6.3	17.8 ± 4.8 *	12.5 ± 1.1	6.0 ± 1.2 *	9.1 ± 1.2	9.6 ± 1.1	8.9 ± 3.1	7.8 ± 1.5	19.2 ± 4.1	21.1 ± 1.6
PHE	70 ± 40	40 ± 20 *	30 ± 10	10 ± 10 *	40 ± 10	50 ± 10	40 ± 30	40 ± 20		
MPFs										
CAT	34.2 ± 10.3	23.7 ± 5.9 *					81.6 ± 20.4	94.7 ± 26.5		
PHE	120 ± 30	230 ± 50 *					90 ± 30	130 ± 40		

**Table 2 ijms-24-07324-t002:** Different enzyme activities in microsomes extracted from roots of near-isogenic tomato plants resistant (Motelle) and susceptible (Moneymaker) to RKNs. Plants were inoculated by *MifieldV* or left uninoculated and used as controls (cntr). Extractions were carried out at 5 dpi. O_2_^−●^ generating activity (NAPDH Ox) was expressed as nmoles cyt. *c* reduced min^−1^ mg^−1^ prot, whereas SOD and isoPEXs as Units mg^−1^ protein. Values are means (*n* = 9) ± SD. Asterisks indicate significant difference with respect to controls according to a paired *t*-test (*p* < 0.05).

	Incompatible Interaction		Compatible Interaction	
	cntr	*+MifieldV*	cntr	*+MifieldV*
NAPDH Ox	15.0 ± 1.7	19.4 ± 2.7 *	13.1 ± 1.7	9.5 ± 1.7 *
SOD	6.7 ± 1.4	15.5 ± 1.9 *	8.6 ± 1.4	11.2 ± 1.1
GUA isoPEX	1.4 ± 0.2	1.7 ± 0.3 *	2.7 ± 0.5	1.9 ± 1.0 *
SYR isoPEX	9.5 ± 1.6	13.6 ± 2.0 *	10.4 ± 1.9	8.7 ± 1.3
PPD-PC isoPEX	8.2 ± 1.2	9.8 ± 2.1	9.9 ± 1.5	8.4 ± 1.1

## Data Availability

Data published in this paper will be available at https://www.researchgate.net/profile/Sergio-Molinari.

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
