# Peer review of "Inhibition of ROS-Scavenging Enzyme System Is a Key Event in Tomato Genetic Resistance against Root-Knot Nematodes"

_ijms, 2023, doi:10.3390/ijms24087324_

Round 1

Reviewer 1 Report

Remark. A more detailed analysis of the problems with explanations is needed:

- assessment of resistance to pathogens, including to certain types of nematodes, for example, with a low content of free and glycosylated SC in Mi-1.2-NahG plants, the latter were infected with a gallic nematode, but these nematodes did not produce eggs (Bhattarai et al., 2008). Thus, the resistance of tomatoes (Lycopersicon esculentum) to M. incognita, M. arenaria and M. javanica is associated with the presence of the dominant gene Mi-1.2. This gene also determines the resistance of tomatoes to aphids (Macrosiphum euphorbiae) and whitefly (Bemisia tabaci) (Milligan et al., 1998; Rossi et al., 1998).

- criteria of resistance, specific (to certain types of nematodes) or non-specific (to all root nematodes), as well as other pathogens (Durrant, Dong, 2004). Along with genetic resistance, determined by the presence of a certain gene(s), another form of plant resistance is being actively studied – induced resistance (IU), which is activated under the influence of phytopathogen metabolites, as well as various factors of biotic and abiotic nature, called elicitors, and reflects a certain adaptive potential of the organism (Durrant, Dong, 2004).

- analysis of the study of the mechanisms of plant resistance to nematodes. It was previously noted that the genetic resistance of plants to nematodes is regulated by salicylic acid using a SC-dependent signaling pathway (Branch et al., 2004; Molinari, Loffredo, 2006; Molinari, 2007), since it is found in most cases with incompatible pathogen-plant combinations controlled by resistance genes (R-genes) (Glazebrook, 2005). It is noted that the accumulation of H2O2 in plant tissues is considered catalase inhibition (Molinari, 2001; Molinari, Loffredo, 2006).

- A taxonomic analysis of pathogens is required, according to the criterion of the degree of pathogenicity, or a reference to research. For example, five species of gallic nematodes have been identified in Eastern Europe – M. incognita (Ko foid & White, 1919) Chitwood, 1949, M. arenaria (Neal, 1889) Chitwood, 1949, M. javanica (Treub, 1885) Chitwood, 1949, M. hapla Chitwood, 1949 and M. ardenensis Santos, 1968. Or a link to the source.

Note to the comments:

Durrant W.E.Dong X. Systemic acquired resistance // Ann. Rev. Phytopathol. 2004. V. 42. â„– 3. P. 185-209.

Bhattarai K.K., Xie Qi-G., Mantelin S. et al. Tomato susceptibility to root-knot nematodes requires an intact jasmonic acid signaling pathway // MPMI. 2008. V. 21. â„–. 9. P. 1205-1214.

Bradford M.M. Rapid and sensitive method for the quantitation of microgram quantities of protein utilizing the principle of protein-dye binding // An. Biochem. 1976. V. 72. â„– 1. P. 248-254.

Branch C.Hwang C.-F.Navarre D.A.Willamson V.M. Salicylic acid is part of the Mi-1-mediated defense response to root-knot nematode in tomato // Mol. Plant-Microbe. 2004. V. 17. â„– 4. P. 351-356.

Cooper W.R.Jia L.Goggin L. Effects of jasmonate-induced defenses on root-knot nematode infection of resistant and susceptible tomato cultivars // J. Chem. Ecol. 2005. V. 31. P. 1953-1967.

Dropkin V. H. The necrotic reaction of tomatoes and other hosts resistant to Meloidogyne: reversal by temperature // Phytopathology. 1969. V. 59. P. 1632-1637.

Farmer E.E.Almeras E.Krisnamurthy V. Jasmonates and related oxylipins in plant responses to pathogenesis and herbivory // Curr. Op. Plant Biol. 2003. V. 6. P. 372-378.

Feussner I.Wasternack C. The lipoxygenase pathway // Annu. Rev. Plant Biol. 2002. V. 53. P. 275-279.  EDN: LUPMDX

Fujimoto T.Tomitaka Y.Abe H. et al. Jasmonic acid signaling pathway of Arabidopsis thaliana is import for root-knot nematode invasion // Nematol. Res. 2011а. V. 41. â„– 1. P. 9-17.

Fujimoto T.Tomitaka Y.Abe H. et al. Expression profile of jasmonic acid-induced genes and induced resistance against the import for root-knot nematode (Meloidogyne incognita) in tomato plants (Solanum lycopersicum) after foliar treatment with methyl jasmonate // J. Plant Physiol. 2011b. V. 168. â„–. 8. P. 1084-1097.  EDN: OARULH

Glazebrook J. Contrasting mechanisms of defense aganst biotrophic and necrotrophic pathogens // Ann. Rev. Phytopathol. 2005. V. 43. P. 205-227.

Gutjahr C.Paszkowski U. Weights in the balance: jasmonic acid and salicylic acid signaling in root-biotroph interactions // Mol. Plant-Microbe. 2009. V. 22. â„– 7. P. 763-772.

Holtzman O. Effects of soil temperature on resistance of tomato to root-knot nematode (Meloidogyne incognita) // Phytopathology. 1965. V. 55. P. 990-992.

Howe G.A.Lightner J.Browse J.Ryan C. A. An octadecanoid pathway mutant (JL5) of tomato is compromised in signaling for defense against insect attack // Plant Cell. 1996. V. 8. P. 2067-2077.

Kessier A.Baldwin I.T. Plant responses to insect herbivory: the emerging molecular analysis // Annu. Rev. Plant Biol. 2002. V. 53. P. 299-328.  EDN: MBSDEB

Kuc J. Development and future direction of inducec systemic resistance in plans // Crop Protection. 2000. V. 18. P. 859-861.  EDN: ALFDVJ

Kunkel B.N.Brooks D.M. Cross-talk between signaling pathways in pathogen defense // Curr. Opin. Plant Biology. 2002. V. 5. P. 325-331.

Li C.Willams M.M.Loh Y.-T. et al. Resistance of cultivated tomato to cell content-feeding herbivores is regulated by the octadecanoid-signaling pathway // Plant Physiol. 2002. V. 130. P. 494-503.

Milligan S.B.Bodeau, J.Yaghoobi, J. et al. The root knot nematode resistance gene Mi from tomato is a member of the leucine zipper, nucleotide binding, leucine rich repeat family of plant genes // Plant Cell. 1998. V. 10. P. 1307-1319.

McConn M.Creelman R. A.Bell E. et al. Jasmonate is essential for insect defense in Arabidopsis // Proc. Natl. Acad. Sci. USA. 1997. V. 94. P. 5473-5477.

McDowell J.M.Dangl J.L. Signal transduction in the plant immune response // Trends Biochem. Sci. 2000. V. 25. P. 79-82.  EDN: AEOQKP

Melillo M.T.Leonetti P.Bongiovanni M. et al. Modulation of reactive oxygen species activities and H2O2 accumulation during compatible and incompatible tomato-root-knot nematode interactions // New Phytol. 2006. V. 170. P. 501-512.

Molinari S. Inhibition of H2O2-degrading enzymes in the response of Mi-bearing tomato to root-knot nematodes and salicylic acid treatment // Nematol. Mediterr. 2001. V. 29. P. 235-239.

Molinari S. New developments in understanding the role of salicylic acid in plant defence // CAB Rev. 2007. V. 67. P. 1-10.  EDN: WQWRBL

Molinari S. Salicylic acid as an elicitor of resistance to root- knot nematodes in tomato // Acta Hort. (ISHS) 2008. V. 789. Р. 119-126.

Molinari S. Natural genetic and induced plant resistance, as a control strategy to plant-parasitic nematodes alternative to pesticides // Plant Cell Rep. 2011. V. 30. P. 311-323.  EDN: ONNCJX

Molinari S.Loffredo E. The role of salicylic acid in defense response of tomato to root-knot nematodes // Physiol. Mol. Plant Pathol. 2006. V. 68. P. 69-78.  EDN: MESVBD

Molinari S.Zacheo G.Bleve-Zacheo T. Effects of nematode infestation on mitochondria isolated from susceptible and resistant tomato roots // Physiol. Mol. Plant Pathol. 1990. V. 37. P. 27-37.

Nandi B.Sukual N.C.Banerjee N. et al. Salicylic acid enhances resistance in cowpea against Meloidogyne incognita // Phytopathol. Medit. 2002. V. 41. P. 39-44.

Nombela G.Williamson V.M.Muniz M. The root-knot nematode resistance gene Mi-1.2 of tomato is responsible for resistance against the white fly Bemisia tabaci // Mol. Plant-Microb. Interact. 2003. V. 16. P. 645-649.

Reymond P.Farmer E.E. Jasmonate and salicilate as global signals for defense gene expression //Curr Opin Plant Biol. 1998. V. 1. â„– 5. P. 404-411.

Author Response

Remark. A more detailed analysis of the problems with explanations is needed:

- assessment of resistance to pathogens, including to certain types of nematodes, for example, with a low content of free and glycosylated SC in Mi-1.2-NahG plants, the latter were infected with a gallic nematode, but these nematodes did not produce eggs (Bhattarai et al., 2008). Thus, the resistance of tomatoes (Lycopersicon esculentum) to M. incognita, M. arenaria and M. javanica is associated with the presence of the dominant gene Mi-1.2. This gene also determines the resistance of tomatoes to aphids (Macrosiphum euphorbiae) and whitefly (Bemisia tabaci) (Milligan et al., 1998; Rossi et al., 1998).

authors: the suggested topic and references have been added in the text

- criteria of resistance, specific (to certain types of nematodes) or non-specific (to all root nematodes), as well as other pathogens (Durrant, Dong, 2004). Along with genetic resistance, determined by the presence of a certain gene(s), another form of plant resistance is being actively studied – induced resistance (IU), which is activated under the influence of phytopathogen metabolites, as well as various factors of biotic and abiotic nature, called elicitors, and reflects a certain adaptive potential of the organism (Durrant, Dong, 2004).

authors: the suggested topic and references have been added in the text

- analysis of the study of the mechanisms of plant resistance to nematodes. It was previously noted that the genetic resistance of plants to nematodes is regulated by salicylic acid using a SC-dependent signaling pathway (Branch et al., 2004; Molinari, Loffredo, 2006; Molinari, 2007), since it is found in most cases with incompatible pathogen-plant combinations controlled by resistance genes (R-genes) (Glazebrook, 2005). It is noted that the accumulation of H2O2 in plant tissues is considered catalase inhibition (Molinari, 2001; Molinari, Loffredo, 2006).

authors: the suggested topic and references have been added in the text

- A taxonomic analysis of pathogens is required, according to the criterion of the degree of pathogenicity, or a reference to research. For example, five species of gallic nematodes have been identified in Eastern Europe – M. incognita (Ko foid & White, 1919) Chitwood, 1949, M. arenaria (Neal, 1889) Chitwood, 1949, M. javanica (Treub, 1885) Chitwood, 1949, M. hapla Chitwood, 1949 and M. ardenensis Santos, 1968. Or a link to the source.

authors: taxonomic analysis has been done in the Section Materials and Methods

Reviewer 2 Report

The manuscript reports important and interesting findings on the state of the ROS-scavenging enzyme system in conferring resistance to RKN. The problem statement and justification of the study are clear. The results are presented clearly and supported by experimental evidence. The authors have also discussed their findings at length and provided critical evaluation and insightful comments. 

However, the authors should check for grammatical errors throughout the manuscript and the formatting (i. short forms in random places and inconsistent, missing full stops, italics for certain words). 

Please remove the hyphen that appears many times in the abstract. 

Please check the statement in Lines 17-21 on the expression and activity of CAT. Is it inhibited or not always inhibited in the resistant tomatoes? The word 'However' can be added to the sentence in Line 19. 

Please write H2O2 correctly with subscript 2 throughout the manuscript.

Figure 1 - Please use contrasting colors/patterns or put the label underneath each graph. 

The title for section 2.4 is a bit general. Better to specify the type/group of genes. 

Line 161-163: Sentence hard to follow. Suggest to re-phrase. 

Figure 5 - please add the labels for the y- and x-axis 

Figure 6B - please add a y-axis label and add the graph label (res/sus) similar to Fig 6A

Figure 7B - please add a y-axis label 

Author Response

The manuscript reports important and interesting findings on the state of the ROS-scavenging enzyme system in conferring resistance to RKN. The problem statement and justification of the study are clear. The results are presented clearly and supported by experimental evidence. The authors have also discussed their findings at length and provided critical evaluation and insightful comments.

However, the authors should check for grammatical errors throughout the manuscript and the formatting (i. short forms in random places and inconsistent, missing full stops, italics for certain words).

done

Please remove the hyphen that appears many times in the abstract.

done

Please check the statement in Lines 17-21 on the expression and activity of CAT. Is it inhibited or not always inhibited in the resistant tomatoes? The word 'However' can be added to the sentence in Line 19.

Expression of CAT gene was markedly or slightly inhibited in 2 resistant cvs after nematode inoculation see Fig. 3

Please write H2O2 correctly with subscript 2 throughout the manuscript.

done

Figure 1 - Please use contrasting colors/patterns or put the label underneath each graph.

 done

The title for section 2.4 is a bit general. Better to specify the type/group of genes.

done

Line 161-163: Sentence hard to follow. Suggest to re-phrase.

unfortunately, the numbers of lines I have don't probably coincide with the ones you mentioned

Figure 5 - please add the labels for the y- and x-axis

done

Figure 6B - please add a y-axis label and add the graph label (res/sus) similar to Fig 6A

done

Figure 7B - please add a y-axis label

done

Reviewer 3 Report

The study of crop genetic resistance and immune system has increased in relevance in recent years, due to their relevance in crop yields, as the authors must already know given their long trajectory in this topic. However, the manuscript needs minor revisions prior to be accepted.

Minor comments:

·        The abstract needs major revisions. Eliminate the hyphens that divided the words in two lines, since with the journal format there are words divided in two (lines 10, 12, 14, 16, 21, 26 and 28). Use subscripts for H2O2 (lines 17, 25 and 26). A period is missing at the end of the sentence in line 15. It is not defined what CAT is for the first time in the abstract.

·        Homogenize figure titles, axis titles and axis numbers in size and format. As well as the figure captions (Figures 1,4 and 8 is in italics and the rest are not). Reduce font sizes in tables 1 and 2.

·        Table 1 is not cited in the text, is it really relevant to the article?

·        Significant difference determination between groups is sometimes based on a p < 0.01 and sometimes on a p < 0.05, why?

·        There is a gap in the text from line 127 to 128 due to a table. Realign the text so this does not occur.

·        Figure 5 is on page 7 and its caption on page 8.

·        Results and discussion are intermingled. Lines 140-148, 237-241 and 261-266 are discussion.

·        Place figure 8 after line 332 (since figure 8b is quoted after the figure).

·        Describe the Hoagland solution composition (line 403).

·        Write a space before /R in primers (lines 548-551).

·        Authors self-cite up to 10 times (20% of total citations). Are there other studies to support this information?

Author Response

The study of crop genetic resistance and immune system has increased in relevance in recent years, due to their relevance in crop yields, as the authors must already know given their long trajectory in this topic. However, the manuscript needs minor revisions prior to be accepted.

Minor comments:

  • The abstract needs major revisions. Eliminate the hyphens that divided the words in two lines, since with the journal format there are words divided in two (lines 10, 12, 14, 16, 21, 26 and 28). Use subscripts for H2O2 (lines 17, 25 and 26). A period is missing at the end of the sentence in line 15. It is not defined what CAT is for the first time in the abstract.

done

  • Homogenize figure titles, axis titles and axis numbers in size and format. As well as the figure captions (Figures 1,4 and 8 is in italics and the rest are not). Reduce font sizes in tables 1 and 2.

done

  • Table 1 is not cited in the text, is it really relevant to the article?

Table 1 is cited in Section 2.3 - it is very important to the article

  • Significant difference determination between groups is sometimes based on a p < 0.01 and sometimes on a p < 0.05, why?

in each experiment, when all means are separated with a significance <0.01 p<0.01 is indicated; when most means are separated by a significance <0.05 and few/one are/is separated by a significance <0.01, then only p<0.05 is indicated

  • There is a gap in the text from line 127 to 128 due to a table. Realign the text so this does not occur.

done

  • Figure 5 is on page 7 and its caption on page 8.

it has been corrected in the revised version

  • Results and discussion are intermingled. Lines 140-148, 237-241 and 261-266 are discussion.

in Results Section only description of the figures/tables and their rationale have been described

  • Place figure 8 after line 332 (since figure 8b is quoted after the figure).

done

  • Describe the Hoagland solution composition (line 403).

done

  • Write a space before /R in primers (lines 548-551).

ok

  • Authors self-cite up to 10 times (20% of total citations). Are there other studies to support this information?

the author Molinari has been working since more than 20 years on topics that specifically regard the biochemical mechanisms of resistant tomato to RKNs, very few other important studies exist on these topics and they were cited